# Cross-sectional evaluation of the multidimensional indicators of psychosocial functioning and its sociodemographic correlates among Indian adults: WHO SAGE Study (2007–2010)

**Apurva Barve** [ID] \*, **Courney S. Thomas Tobin**

Department of Community Health Sciences, University of California, Los Angeles, Los Angeles, California, United State of America

\* apurva.barve@gmail.com

## Abstract

This study examined the relationship between sociodemographic characteristics and psychosocial functioning (PF) among Indian adults. Data (N = 11,230) for this study came from the World Health Organization's SAGE (Longitudinal Study of Global Aging and Adult Health) Wave 1 2007–2010. First, multivariable regression analyses (logistic or linear regression depending on the outcome variable) were run to evaluate whether PF indicators varied by gender after controlling other sociodemographic characteristics. Next, the relationship between sociodemographic characteristics and PF indicators was examined using ordinary least square regression (OLS) models and logistic regression models, separately for men and women. Specifically, the PF indicators, including social indicators of interpersonal relationship difficulty, social connectedness, and personal indicators of sleep, affect, perceived quality of life, and cognition were each regressed on sociodemographic factors. All analyses in the study were cross-sectional in nature and conducted using STATA version 15.1. Overall, the study found significant sociodemographic differences in PF among Indian adults that also varied by gender. As such, social and/or economic disadvantage was associated with poorer PF. However, the results demonstrated that socioeconomic patterns in PF were much more nuanced among women than among men. This study adds to previous research on PF in India and provides new insights into how sociodemographic characteristics shape it. A major research implication of this finding is that inconsistent with assumptions of previous research, an increase in SES is not always linked to proportionate increases in PF among women. The study also makes a compelling case for separately examining multiple non-clinical outcomes of psychosocial health.

**Data Availability Statement:** Data is available online and can be accessed by sending requests

for accessing SAGE Wave 1 data to: https://www.
iipsindia.ac.in/content/SAGE-data

**Funding:** The authors received no specific funding
for this work.

**Competing interests:** The authors have declared
that no competing interests exist.

## Introduction

Life expectancy for the Indian population is on the rise. However, progress in the country's overall health status suffers from widespread sociodemographic disparities that reflect historical and cultural inequities among population subgroups [1, 2]. Similarly, In India the burden of mental and behavioral disorders ranges from 9.5 to 102 per 1000 population and prevalence rate is high in women, elderly, industrial workers, and those having chronic medical conditions [3]. Nearly 7%–13% of disability-adjusted life years (DALYs) worldwide (year 2016) were attributed to mental disorders and India accounted for 15% of these DALYs [4]. Demographic transition, increased life-expectancy, rising chronic diseases, and high unmet health needs in a macro-environment of poverty, stigma, low mental health priority and limited financing, is likely to further accentuate the burden, socio-economic cost and impact of mental disorders in India [4]. Moreover, there is limited understanding of the overall underlying psychosocial functioning (PF) and correlates of mental and physical disorders in India. PF refers to individuals' self-assessment of their social, interpersonal, and personal domains. Existing prospective and cross-sectional research suggests that the multidimensional indicators of PF, such as quality of sleep, level of cognition, quality of interpersonal relationships, affect, are critical building blocks of physical and mental health and demonstrate a social gradient [5]. Nevertheless, the extent to which the distribution of PF is varied across population subgroups, such as women and men, has not been widely examined in the Indian context. Thus, more comprehensive evidence is needed to enhance our understanding of PF and its gender-specific determinants in India. As such, this understanding can help us identify groups at risk for mental and physical disorders.

PF plays an important role in an individual's daily experiences and influences their behaviors, thoughts, and actions. Drawing on the WHO IFC definition of human functioning, the present study operationalizes PF into two main categories: (1) personal psychological functioning and (2) social functioning. Personal functioning is an individual's perception of their functioning in terms of sleep, cognition, negative affect, the quality of life, and experience of depression. Social functioning includes perceived quality of interpersonal relationships and perception of social connectedness with significant others, peers, colleagues, community, and society. This study studies social and personal factors because not only these factors are building blocks of mental health, but these factors are amenable to change and can be potentially targeted in data-driven health policies, interventions, and health improvements programs in India.

There is a need to evaluate the domains of PF collectively, rather than independently. Various indicators of PF (for example, perceived quality of life, social connectedness, depressive symptomology) are commonly studied in the existing literature to evaluate the effects of treatment and interventions, to assess factors that are associated with the development and prognosis of disorders, and to estimate longitudinal recovery and adjustment to acute disorders and/or chronic conditions (for example, [6]). This has led to two important gaps in literature. First, the existing literature on PF is fragmented in nature and these facets of PF have not yet been studied collectively, especially in the Indian public health literature. This prohibits us from understanding the relative impact of each of the facets of PF on individual health. Additionally, there is a reason to believe that the levels of PF and its relationship with sociodemographic correlates are strongly dependent on gender. Indian society is traditionally and historically stratified based on gender. Thus, gender shapes opportunities and vulnerabilities among men and women differently based on socialization processes and structural inequalities [7]. Existing research consistently demonstrates a female-disadvantage in terms of education, income, decision making powers, financial empowerment, and ultimately health in India [8, 9]. Accordingly, one can argue that not only would the levels of PF vary based on gender, but that the

sociodemographic patterns in PF would also be different among men and women. However, there is an overall dearth of literature that explains PF and its relationship to health among women and men in the Indian context. The aim of this study is twofold: first, to examine gender disparities in psychosocial functioning (PF) among the Indian population; and second, to investigate how PF is associated with various sociodemographic factors within distinct subgroups of the Indian population.

## Gaps in literature

Despite the frequent use of PF to assess human health, surprisingly little is known about the social distribution of PF, especially in the Indian context. Furthermore, the study of PF in public health literature has dramatically skewed toward the study of disorders and explanations of social inequalities in health outcomes. As a result, the knowledge of social correlates of PF independent of a disorder diagnosis is scant in the Indian context and also other countries [10]. Lack of information on the PF independent of any disease diagnosis is a key research gap because it prevents our understanding of how and why some of the indicators of PF vary between different subgroups of the population. Furthermore, over-reliance on using the lens of common mental health and physical health diseases to study population wellbeing and overlooking the available psychosocial resources also constrains our comprehensive understanding of protective and risk factors among different population subgroups. Identifying sociodemographic patterning of PF will also highlight the role of social determinants of health in India–opportunities for psychosocial wellbeing are not equally distributed and are highly correlated with social conditions and other life-course related variables, such as, current SES, and other contextual socioeconomic correlates such as the area of residence.

Similarly, to understand the complete impact of PF on health, there is a need to study different facets of PF simultaneously, rather than independently. The evidence to suggest that different facets of PF do not exist in a silo and are influenced by each other. For instance, poor quality of sleep is associated with negative affect and cognitive difficulties [11, 12]. However, there is an overall paucity of research that evaluates collective PF among Indian adults. Aneshensel [13] encourages health researchers to evaluate a full range of theoretically derived outcomes simultaneously to inform our understanding of how social conditions affect overall health. Aneshensel [13] argues that researchers so far have largely used the 'Social Etiological Model' to identify social risk factors related to the occurrence of a particular disorder. The social etiological model often focuses on a single disorder, such as anxiety disorder, and all respondents with the disorder are treated as 'positive' for the outcome under investigation and others are treated as 'negative' on the outcome. However, because of the inherent nature of such a model of investigation, respondents who are not diagnosed with the disorder that is singled out for the investigation are considered 'well'. This leads to the discounting the full impact of a social condition (such as poverty) on wellbeing. Therefore, the goal of the present study is to conduct a comprehensive examination of the relationship between sociodemographic characteristics by examining a full range of relevant indicators of PF.

The Fundamental Cause Theory [14] is particularly relevant and useful in this context. Link & Phelan argue that fundamental social conditions, such as SES or stigma, create health inequities. For example, poverty is widely established as one of the biggest correlates of virtually all health disorders in India and other countries [15, 16]. However, the sociodemographic correlates of PF are less clear in the Indian context. The unique urgency of examining how sociodemographic factors influence PF in India cannot be overstated as almost 363 million Indians (29.5% of India's total population) live below poverty lines and experience wide-spread sociodemographic disadvantages. Evaluation of the sociodemographic correlates of common psychological

disorders and physical health disorders in India is receiving increasing research attention [17]. Yet, many of such studies analyze PF as a side-effect or outcome of a disorder and lack a gender-focused investigation. Without information on social gradient in PF, vulnerable population groups are at risk of being hidden and marginalized, leading to even greater unmet psychosocial needs and disparities among populations. For instance, there is a need to study how sociodemographic indicators, such as education, social caste, religion, influence individuals' PF such as levels of interpersonal relationship and quality of sleep among others. This information is especially essential in India owing to the wide diversity in terms of cultural, linguistic, geographical, infrastructural, and social aspects of lived experiences of populations. Thus, to plan health interventions, guide health policy development, and inform public health research and practice, it is important to know how sociodemographic characteristics influence PF in the Indian context. Therefore, the primary objective of the current study is to analyze the relationships between sociodemographic characteristics and indicators of PF among Indian adults.

## Components of PF and its association with sociodemographic characteristics

**Personal functioning.** *Negative affect*. Negative affect is one of the strongest predictors of psychiatric disorders, particularly depression and anxiety [18]. Borrowing from Watson et al. definition of negative affect, negative affect is operationalized here as general subjective distress that includes a variety of aversive moods such as worry, nervousness, sadness, and loneliness. There is a very limited amount of research to describe how negative affect varies based on sociodemographic characteristics among Indian adults. There is some evidence to suggest that negative affect is higher among women and older adults [19], rural women [20], and among women who were not working as compared to employed Indian women [21]. Nevertheless, some of the key limitations of these studies include a lack of representative sample and lack of psychometrically validated measures.

*Depression*. A bulk of research explains the sociodemographic correlates of depression among Indian adults [17, 22]. Evidence suggests that a greater number of depressive symptoms are associated with older age, low SES, female gender, divorce, bereavement, employment in informal sector, physical diseases, and illiteracy in India [23, 24]. While the urban-rural inequities in physical health are widely documented in India [25], research findings that describes rural/urban disparities in psychosocial health are mixed. Some studies suggest that while rates of depression are slightly higher among urban populations [26], other researchers argue that overall prevalence rates of depressive disorders, such as depression and anxiety, do not differ greatly between urban vs. rural areas in India [27]. Therefore, there is a need to clarify sociodemographic nuances in the experience of depression in India.

*Sleep*. There is ample evidence to demonstrate that sleep is associated with myriad of common health disorders [28]. Gender differences in sleep are also well-established–women experience higher levels of sleep difficulties than men [29]. There is an overall paucity of literature to uncover how social factors, such as marital status, religion, influence sleep quality among Indian population subgroups. Sleep difficulties perhaps represent an unrecognized public health issue facing India and may be a mechanism through which sociodemographic disadvantages are liked to health inequities in India, especially given the rapidly aging of the population and changes in family structure resulting from economic and labor market changes [30]. Thus, the present study takes a closer look at the sociodemographic correlates of sleep among Indian adults.

*Cognition*. Increased cognitive difficulties are considered as some of the common effects of physical and mental disorders [31, 32]. Furthermore, social disadvantage such as,

unemployment [33], experience of emotional abuse and neglect [34], and adverse early life stress and events [35] are linked to increased cognitive difficulties. However, there is a paucity of research to demonstrate the relationship between overall sociodemographic characteristics and cognitive difficulties independent of any physical or psychological disorders, especially among Indian populations. Thus, the present study aims to add to the literature that demonstrates how sociodemographic characteristics influence experiences of cognitive difficulties among Indian men and women.

*Quality of life.* There is strong evidence to demonstrate that quality of life is impacted by SES, race, gender and health status [36]. However, similar to limitations of research on cognition and sleep, research on the quality of life has been focused on evaluating quality of life as an impact of a particular disorder in the Indian context (for example, [37]). The present study is one of the first studies to evaluate the relationship between sociodemographic characteristics and quality of life among Indian adults using a nationally representative sample.

**Social functioning.** *Social connectedness and interpersonal functioning.* There is a clear link between social connection and an individual's health–adults who are socially isolated and less integrated are more likely to die earlier than people who are socially connected [6]. Drawing from prior research, social connectedness is operationalized here as individual's subjective appraisal of interpersonal closeness with social world–civic participation, social participation, and connections with extended family, peers, and friends [38]. It is an individual's perception of their opportunities for interactions with members of their society. Whereas, the interpersonal relationship is as individuals' perceived ease of their close interpersonal interactions. Deeply rooted social hierarchies guide all types of interpersonal relationships in India. The intersectional identity (age, gender, caste, occupation, SES, religion, marital status, location of residence among others) of the individual is one of the key features that shape the nature of interpersonal interactions in the formal as well as informal settings [39]. Because of the social-hierarchical complexities inherent in interpersonal relationships in India, it is important to study whether sociodemographic characteristics of an individual influence their interpersonal relationships and social. However, there is a paucity of research evidence to describe whether sociodemographic subgroups experience varying levels of social connectedness and difficulties or ease in interpersonal relationships.

This study has two aims. The first aim is to evaluate gender differences in the levels of PF in India. Based on previous research, it is hypothesized that women will have a greater burden of difficulties in their PF than men. The second aim is to assess sociodemographic differences in the levels of PF among Indian men and women. It is hypothesized that experience of social or economic disadvantage will be related to lowered levels of personal and social functioning.

## Methods

### Data and sample

Data for this study came from the World Health Organization's SAGE (WHO Longitudinal Study of Global Aging and Adult Health) Wave 1 2007–2010. SAGE Wave 1 is a nationally representative data that covers a broad range of topics [40]. SAGE is representative of the adult population 18 years and older in six states: Assam, Karnataka, Maharashtra, Rajasthan, Uttar Pradesh, and West Bengal. These states were selected from among Indian states with populations of more than five million based on their geographic region and level of economic and human development. Detailed sampling procedures and administration protocol are described elsewhere [40, 41]. SAGE Wave 1 included a total of 11,230 completed interviews: 4,670 interviews with individuals aged 18–49 (3,625 women and 1,045 men) and 6,560 interviews with individuals aged 50-plus (3,256 women and 3,304 men). The data published by WHO is

completely anonymized and authors had no access to any identifying information about individual participants.

## Measures

**Outcome variables.** *Cognition.* Cognition was measured based on a two-item scale that asks the respondents to rate their perceived cognitive difficulty in the last 30 days [40]. The response categories included (1) none, (2) mild, (3) moderate, (4) severe, and (5) extreme/cannot do. Scores ranged from 1 to 10. Higher scores indicated higher levels of cognitive difficulties. Internal validity in terms of Cronbach's alpha ($\alpha$) was 0.83 for the entire population (men = 0.83; women = 0.82).

*Sleep.* Sleep was measured based on a two-item scale that asks the respondents to rate their perceived sleep difficulty in the last 30 days [40]. The response categories included (1) none, (2) mild, (3) moderate, (4) severe, and (5) extreme/cannot do. Scores ranged from 2 to 10. Higher scores indicated higher levels of reported difficulties in sleep. Internal validity in terms of Cronbach's alpha ($\alpha$) was 0.84 for the entire population (men = 0.83; women = 0.84).

*Affect.* Affect was measured based on a two-item scale that asks the respondents to rate their perceived difficulty experienced in terms of affect in the last 30 days [40]. The response categories included none (1) none, (2) mild, (3) moderate, (4) severe, and (5) extreme/cannot do. Scores ranged from 1 to 10. Internal validity in terms of Cronbach's alpha ($\alpha$) was 0.87 for the entire population (men = 0.86; women = 0.87).

*Quality of life.* Quality of life was measured based on 8-item WHOQOL-Bref scale that asks respondents questions about various aspects of daily living [42]. The items ask respondents to indicate their perceived satisfaction with availability of money to meet their basic needs, their perceived energy, satisfaction with their health, themselves, personal relationships, daily living activities, conditions of living places, overall life satisfaction, and overall perceived quality of life. Items are captured with a five-point Likert interval scale with high scores indicating higher quality of life. Internal validity in terms of Cronbach's alpha ($\alpha$) was 0.85 for the entire population (men = 0.85; women = 0.85).

*Depression.* Depression was categorized based on the ICD-10 criteria of mental and behavioral disorders (WHO 1993). Then based on methods used in previous research this study classified responders as a person suffering from 'moderate to severe depression' [17].

*Interpersonal relationships.* Interpersonal relationship was measured based on a four-item scale that asks the respondents to rate their perceived difficulties in interpersonal activities in the last 30 days [40]. Higher scores indicated higher level of perceived difficulties in interpersonal relationship interactions. Internal validity in terms of Cronbach's alpha ($\alpha$) was 0.86 for the entire population (men = 0.86; women = 0.86).

*Social connectedness.* Based on previous research on social connected using the WHO SAGE data [38], social connectedness was measured with a 9 item scale. Five aspects of social interactions were measured: civic participation, social participation, meeting friends and relatives at respondents' place of residence, and visiting friends and family, participating in religious services. Possible scores could range from 0 to 45, with high scores indicating higher level of social connectedness. Internal validity in terms of Cronbach's alpha ($\alpha$) was 0.73 for the entire population (men = 0.76; women = 0.64).

**Sociodemographic correlates.** *Age.* Respondents between 18–24 years of age were coded as (0) Young Adults, respondents between 25–45 years were coded as (1) early-adults, respondents between 46–65 were coded as (2) Middle-aged adults, respondents between 66–80 years were coded as (3) Older-Adults and 80+ year old respondents were coded as (4) Elderly.

*Education.* Respondents' education level was measured by categorizing the last completed grade level where (0) no formal education, (1) for less than secondary school completed (<7 years of schooling), (2) for secondary school completed (7 years of schooling), (3) high school (between and '7–12 years of schooling) and (4) for some college and or post graduate education.

*Income quintiles.* Respondents' income was treated as a categorical variable with five levels representing the (1) Poorest, (2) Poorer, (3) Middle, (4) Richer and (5) Richest in terms of income.

*Caste.* Social caste was coded as a categorical variable with four levels: (1) General Caste, (2) Scheduled Caste, (3) Scheduled Tribe, (4) Other (constituting respondents who do not have a caste or those who belonged to 'other backward castes').

*Marital status.* Marital status was captured as a categorical variable with three groups, (0) Currently Married, (1) Never married, (2) Widowed and (3) Other (separated/divorced).

*Location.* Based on respondents' geographic location, place of residence was categorized into two groups–(1) Rural and (0) Urban.

*Religion.* Religion was be captured as a categorical variable three groups, (1) Hindu and (2) Muslim and (3) to capture all 'Other' religious groups (Jainism, Sikhism, Buddhism, and Christianity).

*Employment status.* Employment status was captured as a categorical variable with four groups: (0) Private Sector, (1) Unemployed, (2) Public Sector, (3) Self-employed, (4) Informal Employment.

**Analytic strategy.** There were three steps in this analysis. First, weighted means and percentages of key study variables were estimated; chi-square tests were used to assess significant men-women differences (Table 1). Next, each PF indicator was regressed on gender to evaluate whether these indicators varied by gender after controlling other sociodemographic characteristics (Table 2). Finally, the relationship between sociodemographic characteristics and PF indicators were examined using ordinary least square regression (OLS) models and logistic regression model, separately for men and women (Table 3). PF indicators, including social indicators of interpersonal relationship difficulty, social connectedness, and personal indicators of sleep, affect, perceived quality of life, and cognition were each regressed on sociodemographic factors. To evaluate the performance of the multivariate logistic models, we assessed the goodness-of-fit using the Hosmer-Lemeshow tests and also examined the discriminative capability by calculating the area under the receiver operating characteristic (ROC) curve. All analyses in the study were cross-sectional in nature and conducted using STATA version 15.1. Using Stata, missing data was handled by omitting the row with the missing values. To account for the multi-stage sampling design of the study, survey weights were applied throughout.

## Results

Table 1 shows the sample descriptive characteristics by gender. Majority of the sample belonged to early adulthood age category (26–45 years; 52.3%). Among men, 27.1% reported having completed less than primary school education (less than 5 years of school); however, 48.7% of the women reported no formal education. A majority of the sample belonged to the 'General caste' category (60%) and reported being currently married (81.7%). Women were more likely to report being widowed (13.1%) than men (3.3%; *p* < .0001). A majority of the sample also reported being Hindu (84.2%) and living in an urban area (74.3%). Men (76.4%) were more likely to live in urban areas than women (72.2%; *p* = .007). Most of the men reported being 'self-employed' (48.3%) followed by informal employment (26.9%), whereas a

**Table 1. Weighted percent distribution of selected demographic characteristics by gender, WHO SAGE India Wave 1 (2007/10).**

| Sociodemographics | Total | | Men | | Women | | Chi-Square value | p |
|---|---|---|---|---|---|---|---|---|
| | *N* | *%* | *N* | *%* | *N* | *%* | | |
| **Age Group** | | | | | | | 1100 | < .0001 |
| 18–25 (Young Adults) | 1,778 | 14.6 | 487 | 10.4 | 1,405 | 18.8 | | |
| 26–45 (Early Adults) | 6,379 | 52.3 | 2,516 | 53.4 | 3,832 | 51.2 | | |
| 46–65 (Middle-Aged Adults) | 3,194 | 26.2 | 1,373 | 29.2 | 1,742 | 23.3 | | |
| 66–80 (Older Adults) | 809 | 6.6 | 326 | 6.9 | 476 | 6.4 | | |
| 80+ (Elderly) | 38 | 0.3 | 7 | 0.2 | 35 | 0.5 | | |
| **Education** | | | | | | | 763.13 | < .0001 |
| College + | 1,036 | 8.5 | 766 | 12.6 | 270 | 4.4 | | |
| High School | 1,781 | 14.6 | 1,164 | 19.2 | 616 | 10.0 | | |
| Secondary School | 1,981 | 16.2 | 1,189 | 19.6 | 792 | 12.9 | | |
| Less than secondary school | 3,115 | 25.5 | 1,644 | 27.1 | 1,471 | 24.0 | | |
| No formal education | 4,285 | 35.1 | 1,299 | 21.4 | 2,986 | 48.7 | | |
| **Income Quintiles** | | | | | | | 3.71 | 0.447 |
| Poorest | 2,464 | 20.3 | 1,233 | 20.5 | 1,231 | 20.2 | | |
| Poorer | 2,573 | 21.2 | 1,258 | 20.9 | 1,315 | 21.6 | | |
| Middle | 2,410 | 19.9 | 1,232 | 20.4 | 1,178 | 19.3 | | |
| Richer | 2,176 | 18.0 | 1,066 | 17.7 | 1,110 | 18.2 | | |
| Richest | 2,497 | 20.6 | 1,238 | 20.5 | 1,260 | 20.7 | | |
| **Caste** | | | | | | | 9.84 | 0.02 |
| General | 6,790 | 60.7 | 3,422 | 60.1 | 3,369 | 61.3 | | |
| Scheduled Caste | 2,141 | 19.1 | 1,147 | 20.2 | 994 | 18.1 | | |
| Scheduled Tribe | 719 | 6.4 | 356 | 6.3 | 363 | 6.6 | | |
| Other | 1,538 | 13.7 | 768 | 13.5 | 769 | 14.0 | | |
| **Marital Status** | | | | | | | 395.14 | < .0001 |
| Married | 9,967 | 81.7 | 5,196 | 85.7 | 4,771 | 77.8 | | |
| Never Married | 1,161 | 9.5 | 649 | 10.7 | 512 | 8.3 | | |
| Widowed | 1,000 | 8.2 | 200 | 3.3 | 800 | 13.1 | | |
| Other | 70 | 0.6 | 19 | 0.3 | 51 | 0.8 | | |
| **Religion** | | | | | | | 0.71 | 0.701 |
| Hindu | 9,456 | 84.2 | 4,777 | 83.6 | 4,679 | 84.8 | | |
| Muslim | 1,386 | 12.3 | 720 | 12.6 | 666 | 12.1 | | |
| Others | 388 | 3.5 | 217 | 3.8 | 171 | 3.1 | | |
| **Location** | | | | | | | 7.22 | 0.007 |
| Urban | 3,133 | 25.7 | 1,430 | 23.6 | 1,703 | 27.8 | | |
| Rural | 9,065 | 74.3 | 4,633 | 76.4 | 4,431 | 72.2 | | |
| **Employment Status** | | | | | | | 3200 | < .0001 |
| Not working | 3,315 | 29.5 | 199 | 4.6 | 3,811 | 55.4 | | |
| Public Sector | 602 | 5.4 | 375 | 8.6 | 137 | 2.0 | | |
| Private Sector | 1,005 | 9.0 | 505 | 11.6 | 426 | 6.2 | | |
| Self-Employed | 3,673 | 32.7 | 2,100 | 48.3 | 1,139 | 16.6 | | |
| Informal Employment | 2,632 | 23.4 | 1,168 | 26.9 | 1,368 | 19.9 | | |

Notes: Weighted frequencies and percentages are shown; chi-square tests were conducted to evaluate the group differences between men and women

majority of women reported 'not working' (55.4%) followed by informal employment (19.9%). There were no significant gender differences based on income quintiles and religion.

Table 2 evaluates whether gender accounts for differences in indicators of PF among Indian adults controlling for other relevant sociodemographic characteristics, including age, education, income quintile, marital status, social caste, religion, employment status, childhood SES, and location of residence. Results show that gender was a significant predictor of interpersonal

**Table 2. Indicators of psychosocial functioning regressed on sociodemographic characteristics WHO SAGE India, Wave 1 (2007).**

| Model | Depression | | Interpersonal Relationship | | Sleep | | Cognition | | Negative Affect | | Social Connectedness | | Quality of Life | |
|---|---|---|---|---|---|---|---|---|---|---|---|---|---|---|
| | _OR_ | _95% CI_ | _b_ | _SE_ | _b_ | _SE_ | _b_ | _SE_ | _b_ | _SE_ | _b_ | _SE_ | _b_ | _SE_ |
| **Unadjusted** | | | | | | | | | | | | | | |
| Gender (Ref.=Men) | | | | | | | | | | | | | | |
| Women | 1.30* | 1.02 - 1.64 | 0.76*** | 0.11 | 0.4*** | 0.05 | 0.53*** | 0.05 | 0.49*** | 0.07 | -4.00*** | 0.24 | -0.62** | 0.18 |
| Intercept | 0.07*** | 0.06 - 0.09 | 5.92*** | 0.86 | 3.12*** | 0.05 | 3.18*** | 0.04 | 3.37*** | 0.05 | 20.45*** | 0.19 | 29.67*** | 0.16 |
| **Adjusted**[a] | | | | | | | | | | | | | | |
| Gender (Ref.=Men) | | | | | | | | | | | | | | |
| Women | 1.27 | 0.97 - 1.65 | 0.55*** | 0.14 | 0.37*** | 0.07 | 0.49*** | 0.07 | 0.45*** | 0.08 | -2.48*** | 0.22 | -0.31 | 0.22 |
| Intercept | 0.02*** | 0.01 - 0.06 | 5.27*** | 0.26 | 2.07*** | 0.16 | 1.93*** | 0.15 | 2.64*** | 0.18 | 18.72*** | 0.72 | 31.16*** | 0.54 |

Notes: Weighted regression results are shown; SE = Standard Errors; OR = Odds Ratios; Logistic or linear regression models were run depending on outcome variables

* = p < .05

** = p < .001

*** = p < .0001

a = all models were adjusted for age, education, income, area of residence, religion, caste, employment status, and marital status.

difficulties, sleep difficulties, cognition, negative affect, and social cohesion (all $p <$ .05). Specifically, Indian women reported statistically higher levels of difficulties in interpersonal relationships, sleep, cognitive, and higher levels of negative affect than Indian men. However, Indian men reported lower levels of social connectedness than Indian women (b = - 2.49; SE = 0.23; $p <$ .0001). The odds of depression and quality of life did not statistically differ between Indian men and women after controlling for sociodemographic characteristics.

The following section presents regression results for each of the personal and social indicator of PF for Indian men and women controlling for all sociodemographic characteristics (refer to Table 3).

With regard to depression, controlling for all other demographic characteristics, among women, higher odds of depression were associated with middle-age (OR: 2.4, 95% CI 1.36–4.23) and older age (OR: 2.16; 95% CI 1.14–4.08), being widowed (OR: 1.85, CI 1.26–2.7), and 'other' marital status (OR: 6.78, CI 2.02–22.77). Interestingly, never being married and belonging to scheduled caste were associated with lower odds of depression among women (all $p <$ .05). Among men, higher odds of depression were associated with older age (OR: 3.41, CI 1.47–7.89), other caste groups (OR: 1.82, CI 1.06–3.12), and widowhood (OR: 3.07, CI 1.35–7.01) (all $p <$ .05). We also assessed the goodness of fit of these models. Overall, both models, for men and women, demonstrated overall goodness of fit as indicated by the Hosmer-Lemeshow test, suggesting no significant discrepancies between observed and expected outcomes across all strata. The overall discriminatory capability of the model, as measured by adjusted ROC curve, was 0.65 for women and 0.63 for men). In terms of interpersonal difficulties, among women, we higher interpersonal difficulties were strongly associated with increase in age, low or lack of education and widowhood, being Muslim, and employment status. Whereas higher income and belonging to 'other' caste groups was associated with lower interpersonal difficulties among women. Among women, around 15% of the variance in interpersonal difficulties is explained by the sociodemographic factors (R-square 0.15). Among men, higher interpersonal difficulties were also strongly associated with increase in age, low income and unemployment. Belonging to scheduled tribes was associated with lower interpersonal difficulties among men (all $p <$ .05). Among men, around 12% of the variance in interpersonal difficulties is explained by the sociodemographic factors (R-square 0.12). With regard to sleep difficulties, among women, there was a strong association between higher sleep difficulties and

**Table 3. Indicators of psychosocial functioning regressed on sociodemographic characteristics WHO SAGE India, Wave 1 (2007).**

| Sociodemographic characteristics | Depression Women Odds Ratio | Depression Women 95% CI | Depression Men Odds Ratio | Depression Men 95% CI | Interpersonal Relationship Women b | Interpersonal Relationship Women 95% CI | Interpersonal Relationship Men b | Interpersonal Relationship Men 95% CI | Sleep Women b | Sleep Women 95% CI | Sleep Men b | Sleep Men 95% CI | Cognition Women b | Cognition Women 95% CI | Cognition Men b | Cognition Men 95% CI | Affect Women b | Affect Women 95% CI | Affect Men b | Affect Men 95% CI | Social Connectedness Women b | Social Connectedness Women 95% CI | Social Connectedness Men b | Social Connectedness Men 95% CI | Quality of Life Women b | Quality of Life Women 95% CI | Quality of Life Men b | Quality of Life Men 95% CI |
|---|---|---|---|---|---|---|---|---|---|---|---|---|---|---|---|---|---|---|---|---|---|---|---|---|---|---|---|---|
| **Age (Ref. = Young Adults(18–24))** | | | | | | | | | | | | | | | | | | | | | | | | | | | | |
| 25–45 (Early Adults) | 1.58 | 0.92–2.7 | 1.26 | 0.52–3.04 | 0.33* | 0.04–0.61 | 0.16 | -0.38–0.7 | 0.5*** | 0.34–0.66 | 0.25* | 0.01–0.5 | 0.6*** | 0.44–0.75 | 0.21 | -0.04–0.47 | 0.67*** | 0.46–0.88 | 0.07 | -0.28–0.42 | 0.29 | -0.26–0.86 | 0.69 | -0.74–2.14 | -1.52*** | -2.04 – -1.01 | -1.17 | -2.52 – 0.17 |
| 46–65 (Middle-Aged Adults) | 2.4** | 1.36–4.23 | 2.17 | 0.94–5 | 1.33*** | 0.95–1.71 | 0.79** | 0.18–1.41 | 1.24*** | 1.04–1.44 | 0.86*** | 0.55–1.17 | 1.48*** | 1.28–1.69 | 0.97*** | 0.67–1.27 | 0.9*** | 0.65–1.16 | 0.61** | 0.23–0.98 | -0.2 | -0.81–0.4 | 0.21 | -1.23–1.65 | -2.78*** | -3.28 – -2.28 | -2.18** | -3.53 – -0.84 |
| 66–80 (Older Adults) | 2.16* | 1.14–4.08 | 3.41** | 1.47–7.89 | 2.3*** | 1.66–2.93 | 2.43*** | 1.73–3.14 | 1.88*** | 1.58–2.18 | 1.84*** | 1.49–2.18 | 2.38*** | 2.04–2.71 | 2.14*** | 1.82–2.45 | 0.81*** | 0.49–1.14 | 1.26*** | 0.86–1.67 | -1.36*** | -2.08 – -0.63 | -1.46* | -2.89 – -0.03 | -4.06*** | -4.75 – -3.37 | -4.38*** | -5.68 – -3.08 |
| 80+ (Elderly) | 2.05 | 0.65–6.45 | 3.3 | 0.53–20.49 | 4.5*** | 3.03–5.97 | 5.9*** | 3.53–8.27 | 2.28*** | 1.48–3.08 | 2.84*** | 1.85–3.84 | 3.56*** | 2.83–4.3 | 3.9*** | 2.83–4.97 | 1.33** | 0.46–2.21 | 1.94** | 0.67–3.21 | -3.4*** | -4.96 – -1.84 | -6.65*** | -9.29 – -4.02 | -6.64*** | -8.58 – -4.69 | -7.16*** | -9.26 – -5.05 |
| **Education (Ref. = College+PostGrad)** | | | | | | | | | | | | | | | | | | | | | | | | | | | | |
| High School | 1.75 | 0.62–4.93 | 1.63 | 0.5–5.32 | 0.09 | -0.38–0.57 | -0.3 | -0.72–0.11 | 0.2 | -0.01–0.42 | 0.11 | -0.12–0.34 | 0.15 | -0.04–0.35 | 0.12 | -0.11–0.37 | 0.44** | 0.18–0.7 | 0.07 | -0.21–0.37 | 0.23 | -0.48–0.95 | -0.93 | -2.28–0.41 | -0.95* | -1.85 – -0.05 | -0.72 | -1.75–0.3 |
| Secondary School | 1.24 | 0.44–3.46 | 1.44 | 0.44–4.72 | 0.11 | -0.34–0.56 | 0.12 | -0.35–0.61 | 0.08 | -0.13–0.3 | 0.34* | 0.07–0.61 | 0.08 | -0.14–0.31 | 0.24 | -0.01–0.5 | 0.43** | 0.14–0.71 | 0.06 | -0.24–0.37 | -0.32 | -1.16–0.52 | -0.85 | -2.12–0.41 | -1.05* | -1.88 – -0.23 | -1.19** | -2.07 – -0.3 |
| Less than secondary school | 1.72 | 0.64–4.66 | 1.14 | 0.32–3.99 | 0.52* | 0.03–1 | 0.43 | -0.03–0.91 | 0.47** | 0.2–0.73 | 0.32* | 0.03–0.62 | 0.49*** | 0.23–0.75 | 0.26 | 0–0.53 | 0.61*** | 0.31–0.92 | 0.02 | -0.26–0.31 | -0.15 | -0.99–0.69 | -2.04** | -3.51 – -0.57 | -1.91*** | -2.82 – -1.01 | -1.05* | -2.07 – -0.02 |
| No formal education | 2.44 | 0.89–6.65 | 1.1 | 0.3–4.01 | 0.72** | 0.2–1.25 | 0.46 | -0.07–1 | 0.39** | 0.11–0.68 | 0.26 | -0.02–0.56 | 0.81*** | 0.52–1.11 | 0.38* | 0.05–0.71 | 0.66*** | 0.33–0.98 | 0.01 | -0.31–0.34 | 0 | -0.86–0.85 | -2.61** | -4.13 – -1.09 | -1.68*** | -2.64 – -0.72 | -1.04* | -2.05 – -0.03 |
| **Income Quintiles (Ref. = Poorest)** | | | | | | | | | | | | | | | | | | | | | | | | | | | | |
| Poorer | 1.02 | 0.74–1.41 | 0.9 | 0.5–1.62 | -0.26 | -0.61–0.08 | -0.53* | -1.03 – -0.02 | -0.19* | -0.37 – -0.01 | -0.15 | -0.48–0.16 | -0.14 | -0.32–0.04 | -0.17 | -0.46–0.12 | -0.26* | -0.47 – -0.05 | -0.03 | -0.33–0.27 | 0.24 | -0.18–0.67 | 0.89* | 0.03–1.75 | 1.02** | 0.55–1.48 | 1.3** | 0.45–2.14 |
| Middle | 1.27 | 0.87–1.85 | 0.91 | 0.43–1.9 | -0.2 | -0.63–0.22 | -0.73** | -1.28 – -0.19 | -0.1 | -0.3–0.1 | -0.52** | -0.83 – -0.21 | 0.03 | -0.18–0.25 | -0.38** | -0.66 – -0.09 | -0.37** | -0.58 – -0.16 | -0.21 | -0.55–0.12 | 0.74** | 0.25–1.23 | 1.77*** | 0.8–2.73 | 0.91** | 0.33–1.49 | 1.94*** | 1.05–2.83 |
| Richer* | 0.71 | 0.46–1.1 | 0.55 | 0.24–1.22 | -0.59*** | -0.98 – -0.2 | -0.69* | -1.24 – -0.15 | -0.2 | -0.41–0.01 | -0.4* | -0.73 – -0.07 | -0.17 | -0.41–0.07 | -0.46** | -0.74 – -0.18 | -0.58*** | -0.79 – -0.37 | -0.38** | -0.68 – -0.07 | 0.84*** | 0.27–1.42 | 2.73*** | 1.79–3.68 | 2.03*** | 1.42–2.63 | 2.76*** | 1.85–3.68 |
| Richest | 1 | 0.67–1.5 | 0.24*** | 0.1–0.56 | -0.69*** | -1.08 – -0.31 | -1.15*** | -1.71 – -0.59 | -0.21 | -0.43–0 | -0.56*** | -0.89 – -0.23 | -0.25* | -0.49 – -0.02 | -0.59*** | -0.9 – -0.28 | -0.67*** | -0.9 – -0.43 | -0.54*** | -0.88 – -0.21 | 0.88*** | 0.25–1.51 | 3.49*** | 2.24–4.74 | 3.15*** | 2.47–3.83 | 4.58*** | 3.53–5.64 |
| **Caste (Ref. = General)** | | | | | | | | | | | | | | | | | | | | | | | | | | | | |
| Scheduled Caste | 1.1 | 0.81–1.5 | 0.93 | 0.51–1.69 | -0.05 | -0.41–0.3 | 0.15 | -0.46–0.78 | -0.05 | -0.21–0.11 | -0.13 | -0.37–0.1 | 0.08 | -0.09–0.26 | 0.08 | -0.16–0.33 | 0.23* | 0–0.46 | 0.14 | -0.09–0.38 | 0.22 | -0.37–0.81 | 1.08* | 0.24–1.91 | -1.14*** | -1.64 – -0.65 | -0.11 | -0.79–0.56 |
| Scheduled Tribe | 0.46** | 0.26–0.81 | 0.33 | 0.1–1 | 0.84*** | 0.47–1.2 | -0.14 | -0.64–0.35 | -0.04 | -0.31–0.22 | -0.15 | -0.57–0.26 | 0.03 | -0.23–0.29 | -0.06 | -0.47–0.12 | -0.29 | -0.71–0.13 | 0.13 | -0.26–0.53 | 1.36** | 0.44–2.29 | 1.4* | 0.16–2.64 | -0.76* | -1.44 – -0.07 | -0.45 | -1.38–0.48 |
| Other | 1.37 | 0.96–1.95 | 1.82* | 1.06–3.12 | 0.1 | -1.31–1.53 | 0.46 | -1.97–2.9 | 0.12 | -0.09–0.34 | 0.79 | -0.45–2.04 | -0.37*** | -0.58 – -0.17 | 0.09 | -0.12–0.71 | 0.28* | 0.06–0.49 | 0.3* | 0.03–0.57 | 2.25*** | 1.64–2.87 | 2.87*** | 1.65–4.1 | -0.46 | -0.96–0.02 | -0.92* | -1.76 – -0.08 |
| **Marital Status (Ref. = Married)** | | | | | | | | | | | | | | | | | | | | | | | | | | | | |
| Never Married | 0.06** | 0.01–0.34 | 2.14 | 0.95–4.78 | -0.12 | -0.51–0.27 | 0 | -0.55–0.36 | -0.18* | -0.35 – -0.01 | | | 0.16 | -0.05–0.37 | 0.08 | -0.16–0.33 | 0.12 | -0.13–0.39 | -0.18 | -0.52–0.15 | 0.69 | -0.09–1.47 | -0.62 | -2.11–0.86 | 0.55 | -0.07–1.19 | -0.4 | -1.68–0.86 |
| Widowed | 1.85** | 1.26–2.7 | 3.07** | 1.35–7.01 | 0.56* | 0.1–1.02 | -0.5* | -0.96 – -0.04 | 0.51*** | 0.23–0.8 | | | 0.32** | 0.11–0.52 | 0.08 | -0.41–0.58 | 0.9*** | 0.62–1.18 | 0.36 | -0.12–0.85 | -0.36* | -0.71 – -0.01 | -0.24 | -1.69–1.19 | -1.56*** | -2.05 – -1.07 | -0.34 | -1.32–0.63 |
| Other | 6.78** | 2.02–22.77 | 0.54 | 0.05–5.77 | 0.05 | -0.59–0.69 | 0.04 | -0.42–0.51 | 0.67 | -0.28–1.64 | | | 0.59 | -0.45–1.65 | 0.19 | -0.99–1.39 | 1.3 | 0–2.6 | 0.89 | -0.45–2.23 | -1.28 | -2.86–0.29 | -0.6 | -3.23–2.01 | -1.28* | -2.37 – -0.18 | 0.46 | -3.39–4.31 |
| **Location (Ref. = Urban)** | | | | | | | | | | | | | | | | | | | | | | | | | | | | |
| Rural | 1.04 | 0.72–1.51 | 0.97 | 0.55–1.72 | | | 0 | -0.34–0.32 | 0.22* | 0.05–0.39 | 0.41** | 0.12–0.71 | 0.26* | 0.04–0.47 | 0.08 | -0.1–0.27 | 0.12 | -0.13–0.39 | 0.41** | 0.17–0.65 | 0.85* | 0.11–1.6 | 1.84*** | 0.88–2.8 | 0.24 | -0.35–0.85 | -0.26* | -1–0.47 |
| **Religion (Ref. = Hindu)** | | | | | | | | | | | | | | | | | | | | | | | | | | | | |
| Muslims | 0.93 | 0.63–1.36 | 1.64 | 0.92–2.92 | | | 0.11 | -0.44–0.66 | 0.21* | 0.01–0.41 | 0.49*** | 0.23–0.64 | 0.43*** | 0.23–0.64 | 0.12 | -0.1–0.35 | 0.2 | -0.06–0.48 | 0.49*** | 0.19–0.79 | -0.35 | -0.89–0.19 | 1.17* | 0.05–2.29 | -1.11** | -1.92 – -0.3 | -1.03 | -1.93 – -0.13 |
| Other | 0.5 | 0.23–1.1 | 0.8 | 0.23–2.82 | | | 0.1 | -0.72–0.93 | -0.02 | -0.36–0.31 | -0.48* | -0.58–0.28 | -0.02 | -0.43–0.37 | 0.14 | -0.41–0.7 | -0.14 | -0.56–0.26 | -0.48* | -0.93 – -0.04 | 0.53 | -0.38–1.46 | 0.19 | -1.04–1.44 | 0.87* | 0.15–1.59 | 0.72 | -0.21–1.66 |
| **Employment Status (Ref. = Private Sector)** | | | | | | | | | | | | | | | | | | | | | | | | | | | | |

*(Continued)*

**Table 3.** (Continued)

| Sociodemographic characteristics | Depression Women | Depression Men | Interpersonal Relationship Women | Interpersonal Relationship Men | Sleep Women | Sleep Men | Cognition Women | Cognition Men | Affect Women | Affect Men | Social Connectedness Women | Social Connectedness Men | Quality of Life Women | Quality of Life Men |
|---|---|---|---|---|---|---|---|---|---|---|---|---|---|---|
| Unemployed | 0.81 (0.51–1.3) | 0.5 (0.14–1.82) | 0.55* (0.12–0.97) | 1.1** (0.33–1.87) | 0.12 (-0.11–0.37) | 0.33 (0–0.68) | 0.23 (-0.03–0.49) | 0.26 (-0.07–0.59) | -0.11 (-0.48–0.25) | -0.06 (-0.5–0.36) | -2.59*** (-3.26– -1.92) | -1.51 (-3.51– -0.48) | 0.47 (-0.19–1.14) | -2.19 (-4.71– -0.31) |
| Public Sector | 0.5 (0.19–1.31) | 1.26 (0.39–4.11) | 0.37 (-0.33–1.08) | 0.19 (-0.28–0.67) | 0.17 (-0.23–0.58) | -0.08 (-0.4–0.23) | 0.29 (-0.13–0.73) | 0.06 (-0.24–0.37) | 0.13 (-0.35–0.62) | -0.27 (-0.62–0.06) | 1.35* (0.13–2.57) | -0.07 (-1.43–1.28) | 0.69 (-0.31–1.69) | 0.52 (-0.33–1.38) |
| Self-Employed | 0.69 (0.39–1.23) | 2.08 (0.88–4.89) | 0.89** (0.39–1.4) | 0.36 (0–0.73) | 0.12 (-0.17–0.42) | 0.06 (-0.18–0.3) | 0.37 (0.07–0.68) | 0.16 (-0.05–0.38) | -0.18 (-0.55–0.19) | -0.07 (-0.36–0.22) | -1.02** (-1.7– -0.33) | 0.79 (-0.17–1.76) | 0.84* (0.04–1.65) | -0.24 (-0.97–0.48) |
| Informal Employment | 0.95 (0.58–1.56) | 2.11 (0.9–4.92) | 0.64* (0.13–1.16) | 0.33 (-0.06–0.73) | 0.45** (0.19–0.71) | 0.17 (-0.09–0.44) | 0.41** (0.12–0.7) | 0.17 (-0.11–0.45) | 0.13 (-0.23–0.5) | 0.17 (-0.14–0.5) | -0.98** (-1.67– -0.3) | 0.79 (-0.24–1.83) | -0.55 (-1.3–0.18) | -0.18 (-0.88–0.5) |
| **Childhood SES (Ref. = Low)** | | | | | | | | | | | | | | |
| High | 0.98 (0.64–1.49) | 0.6 (0.24–1.52) | -0.34* (-0.61– -0.07) | -0.31 (-0.65–0.02) | -0.19* (-0.34– -0.04) | 0.03 (-0.18–0.25) | -0.14* (-0.28–0) | -0.11 (-0.35–0.13) | 0.02 (-0.15–0.19) | 0.05 (-0.24–0.35) | 0.18 (-0.26–0.64) | -0.33 (-1.25–0.58) | 0 (-0.43–0.44) | 0.04 (-0.73–0.81) |
| Intercept | 0.03*** (0–0.11) | 0.02 (0–0.12) | 5.53 (4.76–6.3) | 5.57 (4.85–6.3) | 2.23 (1.82–2.64) | 2.41 (1.92–2.9) | 1.92 (1.5–2.33) | 2.46*** (2.02–2.9) | 2.79*** (2.27–3.31) | 2.82*** (2.29–3.35) | 16.64*** (15.38–17.9) | 17.15*** (14.92–19.38) | 31*** (29.51–32.49) | 30.7*** (28.82–32.58) |

Notes: Weighted regression results are shown; SE = Standard Errors; OR = Odds Ratios Logistic or linear regression models were run depending on outcome variables

* = p < .05

** = p < .001

*** = p < .0001

increase in age (all p < .0001), low (b: 0.39, CI 0.11–0.68) or lack of education (b: 0.47, CI 0.2–0.73), and widowhood (b: 0.51, CI 0.23–0.8), being Muslim (b: 0.21, CI: 0.01–0.41), income, rural area, and informally employed (b: 0.45, CI 0.19–0.71), whereas never being married was associated with lower sleep difficulties. Among women, around 18% of the variance in sleep difficulties is explained by the sociodemographic factors (R-square 0.18). Among men, higher sleep difficulties were strongly associated with increase in age and moderately associated with low education, low income, and belonging to 'other' caste groups (Table 3). Among men, around 14% of the variance in sleep difficulties is explained by the sociodemographic factor (R-square 0.14).

In terms of cognitive difficulties, among women, higher cognitive difficulties were strongly associated with increase in age and lack of education, moderately associated with widowhood, being Muslim, rural area of residence, and informal employment. Conversely, belonging to 'other' caste group was associated with lower cognitive difficulties (b: -0.37, CI -0.58–0.17). Among women, around 24% of the variance in cognitive difficulties is explained by the socio-demographic variables (R-square 0.24). Among men, higher cognitive difficulties were strongly associated with increase in age and moderately associated with lack of education, and income (all $p < .05$). Among men, around 17% of the variance in cognitive difficulties is explained by the sociodemographic variables (R-square 0.17). With regard to negative affect, among women, higher negative affect was strongly associated with increase in age and widowhood, and moderately associated with low education, income and weakly associated with scheduled and other caste. Whereas never being married was associated with lower negative affect among women (b: -0.26, CI -0.51- -0.01). Among women, this model explained 13% variance in the relationship between negative affect and sociodemographic factors (R-square 0.13). Among men, higher sleep difficulties were strongly associated with increase in age, and moderately associated with income, other caste group, being from rural area and being Muslim. Belong to 'other' minority religions was associated with lower negative affect among men (b: -0.48, CI -0.93 - -0.04). Among men, this model explained 11% variance in the relationship between negative affect and sociodemographic factors (R-square 0.11). In terms of social connectedness, among women, older adults and elderly reported lower social connected than young adults (b = -1.26 and -3.4, respectively), low income, widowhood, and, and employment status. Whereas, higher social connectedness was associated with scheduled tribe (b:1.36, CI 0.44–2.29) and other caste (b: 2.5, CI 1.64–2.87), rural area (b: 0.85, CI 0.11–1.6), and working in public sector (b: 1.35, CI 0.13–2.57), among women. Among women, this model explained 13% variance in the relationship between social connected and sociodemographic factors (R-square 0.13). Among men, low social connectedness was strongly associated with being elderly, low or lack of education, income. Whereas higher social connectedness among men was associated with scheduled castes (b: 1.08, CI 0.24–1.91), and tribes (b: 1.4, CI 0.16–2.64), other caste (b: 2.87, CI 1.65–4.1), and living in a rural area (b: 1.84, CI 0.88–2.8; all $p < .05$). Among women, this model explained 15% variance in the relationship between social connected and sociodemographic factors (R-square 0.15). In terms of quality of life, among women, lower quality of life was strongly associated with increase in age, low education, income, belonging to scheduled caste and tribe, widowhood, and 'other' marital status, and being Muslim. On the other hand, higher quality of life was related to belonging to other minority religions (b: 0.87, CI: 0.15–1.59). Among women, this model explained 25% variance in the relationship between social connected and sociodemographic factors (R-square 0.25). Among men, lower quality of life was strongly associated with age, low education, income, other caste group, and being Muslim (all $p < .05$). Among men, this model explained 24% variance in the relationship between social connected and sociodemographic factors (R-square 0.24).

## Discussion

The present study aimed to evaluate sociodemographic patterns in the multidimensional indicators of PF. Drawing from previous research and the fundamental cause theory of health inequities [14], it was hypothesized that sociodemographic factors would shape the levels of personal and social functioning for Indian adults. Overall, the study found support for these theory-driven key hypotheses–there were significant sociodemographic differences in PF among Indian adults. As such, social or economic disadvantage (in terms of age, income, education, employment status, marital status, and religion, caste) was associated with poorer PF. However, the study observed that socioeconomic patterns in PF were much more nuanced among women than among men. There were also interesting patterns created by some types of sociodemographic indicators that were inconsistent with the theory-based expectation, namely caste, and religion. Overall, this study contributes evidence to previous research and theory while also adding new insights into the ways in which sociodemographic characteristics shape PF in the Indian context. These findings also raise several directions for future research.

### Are there sociodemographic patterns in personal functioning among men and women?

The personal functioning included depression, sleep, cognition, negative affect, and quality of life. The study found that as compared to men, women had higher odds of depression, higher sleep and cognitive difficulties and higher negative affect and lower quality of life (refer to Table 2). The following section discusses key sociodemographic patterns in personal functioning separated by gender.

In line with previous research, results indicated age as one of the most robust predictors of psychosocial wellbeing among Indian adults. Accounting for all other demographic factors, an increase in the age was linked to poorer levels of personal functioning including sleep, cognition, affect, and quality of life for women as well as men. However, the relationship between age and depression was not consistent across all age categories for men and women. For example, odds of depression for middle-aged adults (aged 46–65 years) were higher for women but not for men. Among men, odds of depression were higher only for older adults between the ages of 66–80 years as compared to young men (18–24 years). This indicates that middle-age group was a vulnerable period for depression among Indian women but not for men. These results are consistent with previous studies that demonstrate gender gaps in depression based on stages of life-cycle [43]. While the research on gender-specific geriatric depression in India is growing [44, 45], there is also a need to closely examine the mechanisms that increase risks of depression among middle-aged women. Because depression is also related to various other mental and physical health co-morbidities, middle-aged women are particularly vulnerable groups [46, 47]. Therefore, mental health interventions should target women from this vulnerable age groups.

Key gender differences observed in present study could be attributed to sociopolitical, cultural, and economic inequalities in the lived experiences in India. Some scholars argue that throughout the life-course, relative significance and burden of work and family circumstances vary greatly for men and women in ways that create steep disparities in experiences of stress, strain, financial and economic status, personal autonomy, social authority, and social recognition based on gendered norms and expectations [43, 48]. For instance, roles and responsibilities of child-rearing, upkeep of other important household responsibilities, gaps in pay and discrimination in work opportunities, caring for elderlies, and conflicting work/life balance demands are greater for women than for men [49, 50]. Thus, psychologically, gender differences in middle-aged adults favor men's wellbeing over women [43]. For instance, this study

demonstrated that in Indian context, these gender differences in experience of depression among middle-aged adults remain even after adjusting for other important sociodemographic characteristics.

In terms of SES, low or lack of education was consistently related to sleep, cognitive, and negative affect among women but not among men. Whereas low levels of income were consistently related to difficulties in sleep and cognitive and negative affect among men but not among women. Thus, income created more nuances in personal functioning for men and educated created more gradations in personal functioning for women. This suggests that patterns in personal functioning depend on the type of SES measure (income and education) and gender. Similarly, men who reported at least some exposure to formal education reported no greater difficulties in cognitive functioning than college-educated men. This suggests that the level of education is a protective factor against cognitive difficulties for men even at very low levels of education. Whereas among Indian women, education was a protective factor only after achieving at least middle levels of education (secondary school completion). This was consistent with a study that found that education was a protective factor against deteriorations in cognitive functioning only at middle and higher levels of education among Chinese adults [51]. Thus, although the education is one of the most robust predictors of cognitive functioning [52], the ways in which education offers protective benefits against cognitive functioning differs in important ways based on gender among Indian adults.

Regarding social minority statuses in terms of caste and religion, belonging to 'Other' caste was related to higher odds of depression, higher negative affect, higher sleep difficulties for men but not for women. Interestingly, as compared to general caste population, belonging to historically oppressed caste categories, such as scheduled caste or scheduled tribe, did not create increased difficulties in sleep, cognition, and negative affect for men or women. On the contrary, belonging to scheduled tribe was a protective factor against depression for women. Similarly, while Muslim women reported high sleep and cognitive difficulties as compared to Hindu women, there was no differences in personal functioning in other religious minorities and their Hindu counterparts for men as well as women. Thus, belonging to a historically and politically disadvantaged caste group did not always create lower levels of personal functioning among men and women. On the contrary, women from scheduled tribes and men from 'Other' castes reported lower odds of depression and higher levels of cognitive functioning, respectively. Thus, perhaps caste membership shapes the access and availability of different psychosocial resources, such as social support, beliefs of mastery, coping, that create differences in the indicators of personal functioning. Moreover, the availability of psychosocial resources may not always be linked to the caste hierarchy. On the contrary, men and women from 'Other' religious minorities reported a higher quality of life and lower levels of negative affect than their Hindu counterparts, respectively. Similar to results related to social caste, belonging to minority religion did not unequivocally present detrimental effects on personal functioning among men and women. The results highlight that even though Islam, Jainism, Sikhism, and Christianity are all minority religions in India, only Muslims reported poorer personal functioning. Whereas personal functioning was not different and, in some cases, better for other minority religions than the majority religion group (Hinduism) in India.

Among women, widowhood was detrimental to all indicators of personal functioning as compared to married women. Likewise, being separated or divorced increased women's experience of depression and decreased their quality of life. While widowhood also increased odds of depression for men, overall marital status was not a robust predictor of personal functioning among Indian men. The current study highlights the particularly detrimental effects of widowhood on Indian women. Even after adjusting for all other relevant sociodemographic factors, widowhood emerged as a robust predictor of all indicators of personal and social functioning

among women and increased odds of depression among men. These results are consistent with other research that explored widowhood as a predictor of diseases among Indian adults [53]. Various explanations, such as loss of economic, social, and psychological benefits offered by being in a married relationship, loss of social support, and social status of damaging health effects of widowhood have been provided in the literature [54].

The type of employment was not related to personal functioning among men. Among women, informal employment was related to higher levels of sleep and cognitive difficulties than employment in the private sector. In the present study, around 20% of the total women sample was engaged in informal employment. The informal employment is defined as employment that lacks in the contract-based wage and non-wage related benefits and hence offers little to no long-term financial security [55]. Structural discriminations, poor working conditions, insufficient wages are inherent in informal employment. The study highlights increased psychosocial vulnerabilities related to informal employment among women even after accounting for other SES factors and demographic characteristics. Therefore, evaluating employment in a bivariate manner (employed or unemployed) may lead to erroneous assumptions about the role of employment on individual wellbeing, especially among women in India.

## What are the sociodemographic patterns in social functioning among men and women?

The social functioning included interpersonal relationship (i.e., their relationship with their immediate family) and social connectedness (i.e., connectedness with their larger societal network). The study found that as controlling for all sociodemographic covariates, compared to men, women had higher interpersonal relationship difficulties and lower social connectedness (refer to Table 2). The following section discusses key sociodemographic patterns in social functioning separated by gender.

The findings revealed that as men and women age their difficulties in interpersonal functioning increase, however, the perceived levels of social connectedness decrease only after the age of 65 years, among Indian men and women. These findings show the unique relationship between an individual's age and the level of social connectedness and the perceived quality of interpersonal relationships. Consistent with previous research [56, 57], even though the frequency of social connectedness does not decrease before the age of 65 years for men and women, the perceived ease of interpersonal functioning is diminished as Indian men and women age. This is consistent with research that suggests that age is negatively related to social network size, closeness to network members, frequency of contact, and the number of non-primary-group ties. Interestingly among men, as compared to young adult men, difficulties in the interpersonal relationships do not increase until the age of 46 years. However, difficulties in the interpersonal relationships increase for women at every stage of age in a linear fashion. As such, the present study adds to the growing literature of the unique relationship between different stages of age, gender, and PF. Future studies should explore the relationship between age-related linear decrease in interpersonal relationship quality and other aspects of psychosocial functioning among Indian women.

Low levels of education (namely, no formal education and less than primary school education) were related to higher interpersonal difficulties among women and lower levels of social connectedness among men. However, middle to higher levels of education did not influence social functioning among men and women. Thus, the results highlight that social functioning does not differ across educational levels in the Indian context once men and women have achieved a certain threshold of education (secondary school completion). Similarly, in line with previous research [58] and fundamental cause theory, income was positively related to

social functioning among Indian men in a linear fashion. However, among women only those belonging to richer and the richest income quintiles reported lower levels of interpersonal difficulties than women from poorest income quintiles. Taken together, this indicates that any increase in income is linked to higher levels of social functioning among Indian men; however, Indian women receive these protective benefits of income only after reaching a certain threshold of income, namely middle-income quintile for social connectedness and richer income quintile for interpersonal interactions. There is a need to clarify the mechanisms that create these important gradations among women.

Contrary to the hypotheses, women from 'Other' caste groups and men from scheduled tribe reported lower levels of difficulties in their interpersonal relationship than their general caste counterparts. Likewise, social connectedness was higher among women from scheduled tribes and 'Other' castes and men from scheduled castes, scheduled tribes, and 'Other' caste groups than their general caste counterparts. Taken together, the present results indicate that historical disadvantages in terms of social caste hierarchy do not always translate into disadvantages in terms of interpersonal relationships and overall social functioning. In reality, social functioning was not different or lower among historically disadvantaged caste groups than hierarchically advantaged social castes. Further research is urgently required to uncover the ways in which health disparities are created based on social castes in India.

Caste identity is one of the most pervasive forms of social stratification and inequities in Indian society. However, making a global assumption that caste disadvantages are also unequivocally linked to disadvantages in terms of personal and social functioning would be erroneous. There is a need to investigate the role of the social, political, structural, and cultural discrimination inherent in the social caste system as one of the determinants of functioning and health. The present study shows that although membership in a disadvantaged caste group created disparities in some facets of personal functioning, it served as a protective factor for other indicators of personal and social functioning.

Similar to the association between personal functioning and type of employment, women who were informally employed also indicated more difficulties in interpersonal relationships and lowered levels of social connectedness than women who worked in the private sector. Among men, unemployment was related to increased difficulties in interpersonal relationships. However, no other indicators of social functioning differed based on the type of employment among men. Thus, the present research emphasizes the increased vulnerability of informally employed women for poor social and personal functioning.

## Limitations

A few important caveats limit the conclusions of the study. First, gender is reported as a binary variable (male and female) in WHO SAGE Wave 1. This restricts the generalizability of the findings to include all categories of gender and sexual identities. Second, the cross-sectional nature of the study precludes causal conclusions. However, prior research and theory suggests that social structures, social hierarchy, and social conditions shape individuals' life experiences, opportunities and accessibility of resources that perpetuate disparities in functioning and well-being. Future research should evaluate changes in sociodemographic characteristics, such as social mobility, and it impact on PF, using longitudinal data. Third, the data was collected over a 13 years ago and given the changes in the Indian society in the last decade, it would be beneficial to update the findings with the latest data from WHO SAGE Wave 2 India data. Meanwhile, the findings from this study can still provide valuable insights into the relationship between psychosocial functioning and sociodemographic characteristics. Future research using newer datasets that highlights differences in health outcomes especially as they relate to

gender and other socioeconomic variables such as caste is required. Finally, health behaviors are an important mechanism through which sociodemographic factors create health impacts. However, behaviors are not included in the study. Future studies should evaluate whether health behaviors, such as physical activity, diet, and substance use influence the relationship between sociodemographic variables and PF among these populations. We also recognize that the potential for measurement error in this study could introduce potential limitations to the findings. However, the instruments used in this study were validated by the WHO SAGE survey team [42].

## Conclusion and implications

Findings from this study show that there are distinct sociodemographic patterns in PF across gender groups. While the relationship between sociodemographic characteristics and PF is in line with hypotheses derived from fundamental cause theory and other sociological theories of inequities based on stratification and hierarchy, the present study observed patterns that are inconsistent with broader theoretical expectations. For instance, based on theory, one would argue that disadvantaged status will always confer disadvantages in PF. Nonetheless, the present study found that some of the historically disadvantaged groups (e.g. lower caste groups and religious minorities) reported higher levels of functioning than the hierarchically advantaged groups. As such, results demonstrate that sociodemographic patterns in PF depend on the psychosocial indicator under investigation. This finding implies that future research needs to clarify the underlying mechanisms that create widespread inequities in health and wellbeing among Indian adults. For example, future research should study structural, political, cultural, and personal discrimination faced on different caste and religious groups as one of the fundamental causes of inequities in India. Current explanations of inequities based on mere membership into castes or religions are inadequate.

This study also offers insights demonstrating noteworthy gender-based nuances in sociodemographic patterns for PF. While the study observed the gradation in levels of PF depending on sociodemographic features, the protectiveness of sociodemographic advantages was not equivalent across both genders. The links between sociodemographic characteristics and different facets of PF are much more nuanced for Indian women than for Indian men. In other words, the relative importance of sociodemographic characteristics depends on the type of PF and the gender of the respondent.

A major research implication of this finding is that inconsistent with assumptions of previous research, an increase in SES is not always linked to proportionate increases in PF among women. For instance, women in the present study needed to reach a higher threshold of income and education to receive the protective benefits of these statuses. The findings in this context are consistent with 'the diminishing returns' research that explains distinctive racial patterns in health returns of income and education in the United States [59, 60]. This result has important public health implications as it suggests that public health programs need to take a gender-focused approach to tailor interventions to address psychosocial health needs among these populations. Similarly, future research in this area need to explore mechanisms related to creation of gender-based disparities in this population.

Furthermore, the present study makes a compelling case for examining multiple outcomes of psychosocial health. For instance, the study observed that even though some of the socioeconomic indicators (education and income) did not predict the likelihood of depression among men and women, they had clear links to negative affect among women. Negative affect is one of the strongest predictors of psychological disorders [18]. This has important implications for future research–studying only the disease outcome (depression) and ignoring indicators of PF

would lead to erroneous assumptions about the importance of sociodemographic features that are related to wellbeing.

Finally, the findings imply that public health programs should focus on increasing holistic PF among Indian population, especially among population subgroups that are especially vulnerable to adverse PF given sociocultural, political, and historical factors. Future research should also study the multiple vulnerable subgroups identified by the present study, including older and elderly adults, widowed individuals, women working in informal sections, rural populations, and Muslim men and women.

## Supporting information

**S1 Checklist. STROBE statement—checklist of items that should be included in reports of observational studies.**
(DOCX)

## Author Contributions

**Conceptualization:** Apurva Barve, Courney S. Thomas Tobin.

**Data curation:** Apurva Barve.

**Formal analysis:** Apurva Barve.

**Investigation:** Apurva Barve.

**Methodology:** Apurva Barve, Courney S. Thomas Tobin.

**Software:** Courney S. Thomas Tobin.

**Supervision:** Courney S. Thomas Tobin.

**Validation:** Apurva Barve, Courney S. Thomas Tobin.

**Writing – original draft:** Apurva Barve.

**Writing – review & editing:** Courney S. Thomas Tobin.

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
