## [Decision Letter · Decision Letter 0]

2 Nov 2023

PGPH-D-23-01132

Cross-sectional evaluation of the multidimensional indicators of psychosocial functioning and its sociodemographic correlates among Indian adults: WHO SAGE Study (2007-2010)

Dear Dr. Barve,

Thank you for submitting your manuscript to PLOS Global Public Health. After careful consideration, we feel that it has merit but does not fully meet PLOS Global Public Health’s publication criteria as it currently stands. Therefore, we invite you to submit a revised version of the manuscript that addresses the points raised during the review process.

We look forward to receiving your revised manuscript.

Kind regards,

Daniel Kim, M.D., Dr.P.H.

Academic Editor

Journal Requirements:

Additional Editor Comments (if provided):

Reviewers' comments:

Reviewer's Responses to Questions

**Comments to the Author**

1. Does this manuscript meet PLOS Global Public Health’s publication criteria? Is the manuscript technically sound, and do the data support the conclusions? The manuscript must describe methodologically and ethically rigorous research with conclusions that are appropriately drawn based on the data presented.

Reviewer #1: Partly

Reviewer #2: Yes

2. Has the statistical analysis been performed appropriately and rigorously?

Reviewer #1: Yes

Reviewer #2: Yes

3. Have the authors made all data underlying the findings in their manuscript fully available (please refer to the Data Availability Statement at the start of the manuscript PDF file)?

Reviewer #1: Yes

Reviewer #2: Yes

4. Is the manuscript presented in an intelligible fashion and written in standard English?

Reviewer #1: Yes

Reviewer #2: Yes

5. Review Comments to the Author

Reviewer #1: The article needs following clarifications:

1. The work was conducted and completed nearly 13 years back i.e. 2007 – 2010. Thus the relevance of the findings in the present scenario is doubtful.

2. Please check the headings of the manuscript as per journal norm i.e. ‘Background’ under the heading ‘Introduction’.

3. Line: 290 – 291: Under the heading of 'affect' – it is written as sleep was measured.........Please correct.

4. Better to mention the chi-square statistics along with P value in the table.

5. For multivariable logistic regression analysis, please mention the model fitness, variation of dependent variable that can be explained from independent variables, proportion of dependable variable explained etc. as a running text in the result section.

6. Table 1, 2, 3: Stub heading is missing.

Reviewer #2: Overall this is a strong manuscript that makes useful contributions to the literature on social determinants of psychosocial health and mental health in India. The analyses are thoughtfully done and findings have implications for research and practice. With minor revisions, I believe it would be appropriate for publication in PLOS Global Public Health.

6. PLOS authors have the option to publish the peer review history of their article (what does this mean?). If published, this will include your full peer review and any attached files.

**Do you want your identity to be public for this peer review?** For information about this choice, including consent withdrawal, please see our Privacy Policy.

Reviewer #1: No

Reviewer #2: **Yes: **Khushbu Balsara

---

## [Decision Letter · Decision Letter 1]

16 Jan 2024

PGPH-D-23-01132R1

Cross-sectional evaluation of the multidimensional indicators of psychosocial functioning and its sociodemographic correlates among Indian adults: WHO SAGE Study (2007-2010)

Dear Dr. Barve,

Thank you for submitting your manuscript to PLOS Global Public Health. After careful consideration, we feel that it has merit but does not fully meet PLOS Global Public Health’s publication criteria as it currently stands. Therefore, we invite you to submit a revised version of the manuscript that addresses the points raised during the review process.

We look forward to receiving your revised manuscript.

Kind regards,

Daniel Kim, M.D., Dr.P.H.

Academic Editor

Journal Requirements:

2. We have noticed that you have a list of Supporting Information legends in your manuscript. However, there are no corresponding files uploaded to the submission. Please upload them as separate files with the item type 'Supporting Information'. 

Additional Editor Comments (if provided):

The authors should add some comment on the use of data from Wave 1 in 2007-2010 as opposed to Waves 2 or 3 and any related implications.

Reviewers' comments:

Reviewer's Responses to Questions

**Comments to the Author**

1. If the authors have adequately addressed your comments raised in a previous round of review and you feel that this manuscript is now acceptable for publication, you may indicate that here to bypass the “Comments to the Author” section, enter your conflict of interest statement in the “Confidential to Editor” section, and submit your "Accept" recommendation.

Reviewer #2: All comments have been addressed

2. Does this manuscript meet PLOS Global Public Health’s publication criteria? Is the manuscript technically sound, and do the data support the conclusions? The manuscript must describe methodologically and ethically rigorous research with conclusions that are appropriately drawn based on the data presented.

Reviewer #2: Yes

3. Has the statistical analysis been performed appropriately and rigorously?

Reviewer #2: Yes

4. Have the authors made all data underlying the findings in their manuscript fully available (please refer to the Data Availability Statement at the start of the manuscript PDF file)?

Reviewer #2: Yes

5. Is the manuscript presented in an intelligible fashion and written in standard English?

Reviewer #2: Yes

6. Review Comments to the Author

Reviewer #2: All comments have been addressed.

7. PLOS authors have the option to publish the peer review history of their article (what does this mean?). If published, this will include your full peer review and any attached files.

**Do you want your identity to be public for this peer review?** For information about this choice, including consent withdrawal, please see our Privacy Policy.

Reviewer #2: **Yes: **Khushbu Balsara

---

## [Editor Report · Decision Letter 2]

25 Mar 2024

Cross-sectional evaluation of the multidimensional indicators of psychosocial functioning and its sociodemographic correlates among Indian adults: WHO SAGE Study (2007-2010)

PGPH-D-23-01132R2

Dear Dr Barve,

We are pleased to inform you that your manuscript 'Cross-sectional evaluation of the multidimensional indicators of psychosocial functioning and its sociodemographic correlates among Indian adults: WHO SAGE Study (2007-2010)' has been provisionally accepted for publication in PLOS Global Public Health.

Best regards,

Daniel Kim, M.D., Dr.P.H.

Academic Editor